# Randomised controlled trial combining vitamin E-functionalised chocolate with physical exercise to reduce the risk of protein–energy malnutrition in predementia aged people: study protocol for Choko-Age

Anna Pedrinolla,[1] Masoud Isanejad [ID],[2] Cinzia Antognelli,[3] Desirée Bartolini,[4] Consuelo Borras,[5] Valentina Cavedon,[6] Gabriele Di Sante,[6] Anna Migni,[7] Cristina Mas-Bargues,[8] Chiara Milanese,[6] Claudia Baschirotto,[6] Roberto Modena,[9] Alessandra Pistilli,[6] Mario Rende,[6] Federico Schena,[6] Anna Maria Stabile,[6] Nicola Vincenzo Telesa,[3] Sara Tortorella [ID],[10] Kay Hemmings,[2] Jose Vina,[5] Eivind Wang,[9,11] Anne McArdle,[2] Malcolm J Jackson,[2] Massimo Venturelli,[6] Francesco Galli[12]

For numbered affiliations see end of article.

**Correspondence to**
Dr Masoud Isanejad;
masoudi@liverpool.ac.uk

## ABSTRACT

**Objective** Protein–energy malnutrition and the subsequent muscle wasting (sarcopenia) are common ageing complications. It is knowing to be also associated with dementia. Our programme will test the cytoprotective functions of vitamin E combined with the cortisol-lowering effect of chocolate polyphenols (PP), in combination with muscle anabolic effect of adequate dietary protein intake and physical exercise to prevent the age-dependent decline of muscle mass and its key underpinning mechanisms including mitochondrial function, and nutrient metabolism in muscle in the elderly.

**Methods and analysis** In 2020, a 6-month double-blind randomised controlled trial in 75 predementia older people was launched to prevent muscle mass loss, in respond to the 'Joint Programming Initiative A healthy diet for a healthy life'. In the run-in phase, participants will be stabilised on a protein-rich diet (0.9–1.0 g protein/kg ideal body weight/day) and physical exercise programme (high-intensity interval training specifically developed for these subjects). Subsequently, they will be randomised into three groups (1:1:1). The study arms will have a similar isocaloric diet and follow a similar physical exercise programme. Control group (n=25) will maintain the baseline diet; intervention groups will consume either 30 g/day of dark chocolate containing 500 mg total PP (corresponding to 60 mg epicatechin) and 100 mg vitamin E (as RRR-alpha-tocopherol) (n=25); or the high polyphenol chocolate without additional vitamin E (n=25). Muscle mass will be the primary endpoint. Other outcomes are neurocognitive status and previously identified biomolecular indices of frailty in predementia patients. Muscle biopsies will be collected to assess myocyte contraction and mitochondrial metabolism. Blood and plasma samples will be analysed for laboratory endpoints including nutrition metabolism and omics.

## STRENGTHS AND LIMITATIONS OF THIS STUDY

⇒ A novel polyphenol and vitamin E combined with exercise intervention programme in predementia to improve muscle health.
⇒ The programme is designed to improve protein–energy malnutrition, muscle mass, functional and cognitive abilities.
⇒ The study enables the acquisition of high-quality biological human data to inform the mechanism of muscle mass decline in older people. The study and its results are assumed to be applicable among older people as an effective intervention strategy.
⇒ A limitation to this study was, those unable to follow the exercise programme due to disability were not included to the study.
⇒ The study did not have prior patient and public involvement.

**Ethics and dissemination** All the ethical and regulatory approvals have been obtained by the ethical committees of the Azienda Ospedaliera Universitaria Integrata of Verona with respect to scientific content and compliance with applicable research and human subjects' regulation. Given the broader interest of the society toward undernutrition in the elderly, we identify four main target audiences for our research activity: national and local health systems, both internal and external to the project; targeted population (the elderly); general public; and academia. These activities include scientific workshops, public health awareness campaigns, project dedicated website and publication is scientific peer-review journals.

**Trial registration number** NCT05343611.

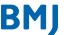

## INTRODUCTION

Protein–energy malnutrition (PEM) is a condition caused by insufficient intake of protein and energy, and it is a significant concern for older adults. Current data shows that 23% of European older adults are at high risk and 48% at *some* risk of PEM.[1] PEM is a significant risk factor for loss of muscle mass, physical function (ie, sarcopenia), frailty[1–3] and cognitive decline.[4 5]

Cognitive decline and dementia are commonly linked to a decrease in muscle mass and physical function in older adults. This may be caused by an abnormal regulation of the hypothalamic–pituitary–adrenal (HPA) axis, leading to an alteration of glucose production, and the reduction in energy supply to the muscles[6]; the chronically elevated cortisol level[7 8]; and impaired energy and tissue anabolism pathways, cellular redox and reactive oxygen species (ROS) signalling, accumulatively increasing tissue degeneration in older people.[9] Other possible mechanisms of tissue damage are defective modulation of genes involved in immunoinflammatory pathways causing low-grade inflammation[10] and hypercortisolism.[8]

Combined nutrition and physical exercise interventions are considered a promising strategy to counteract PEM, muscle mass and physical function loss in older age,[3 5] but the optimal protocols required are yet to be determined. Dietary protein is a prerequisite for the maintenance of skeletal muscle mass; stimulating muscle protein synthesis, and it has been shown to overcome anabolic resistance and muscle mass preservation in malnourished older individuals.[11] Interestingly, vitamin E supplementation improves myotube survival and prevents myoblasts from atrophy, also stimulating their proliferation and promoting membrane repair during exposure to oxidant challenges which attenuate senescence and replenish the regenerative capacity of these cells.[12 13] Further, polyphenols (PP) are associated with inhibition of inflammatory cytokines, oxidative stress and muscle atrophy-related ubiquitin ligases.[14 15] Also, PP support the activation of insulin growth factor-1 (IGF-1) signalling pathway, mitochondrial biogenesis and function and differentiation factors involved in myogenesis.[14]

A nutritional intervention to counteract PEM should preferably be combined with physical exercise. Previous randomised controlled trials (RCTs) have shown the muscle anabolic exercise effect on muscle mass and strength[16] and on the HPA axis reactivity and attenuation of hypercortisolism in patients with Alzheimer's disease.[17] Particularly, the maximal strength training (MST) has previously yielded excellent results in older subjects.[18] Also, the high-intensity interval training (HIIT) was previously shown to be excellent for improving health and performance in older subjects[19] and frail patient populations.[20] These results suggest that concurrent interventions consisting of both MST and HIIT can induce complementary, beneficial effects on muscle strength, mass and metabolism.[21] The effect of PP or vitamin E combined with effective HIIT to improve muscle metabolism has not been investigated. Some animal studies have provided evidence that the combination of exercise, PP and vitamins (ie, vitamin C and E) improved cellular antioxidant enzyme contents and reduced ROS levels.[22 23] However, most of the findings obtained from in vitro studies and animal models have not been confirmed or fully investigated in clinical trials.

The main hypothesis is the antioxidant and cytoprotective functions of vitamin E,[24] combined with the cortisol-lowering effect of chocolate PP,[25] can prevent the age-dependent decline of mitochondrial function and nutrient metabolism in skeletal muscle. We will test this hypothesis in a group of older people with stabilised dietary protein intake and physical activity levels. This is important as we are able to account for these two major lifestyle factors and age-dependent muscle function decline before testing the metabolic effects of chocolate PP and vitamin E. The RCT protocol encompasses a wide range of clinical and molecular investigations of skeletal muscle mitochondrial function, nutrient metabolism, insulin and cortisol signalling, omics and tracing the enzymatic and free radical-derived metabolites of vitamin E for the first time in the human muscle.

## METHODS
### Study design

This is a multicentre study, screening for eligible participants will be from four centres, Section of Movement Science of the Department of Neuroscience, Biomedicine, and Movement Science of University of Verona, Italy; Centro di Ricerca Sport Montagna e Salute—Rovereto, Italy; P. Pederzoli Hospital—Peschiera, Italy; and Mons. Mazzali Foundation—Mantua, Italy. The study has been registered with ClinicalTrials.gov Identifier: NCT05343611.

### Inclusion criteria

Inclusion criteria include the presence of mild cognitive impairment or mild dementia. Recruited individuals will be assessed by means of neuropsychological tests (Mini–Mental State Examination (MMSE), evaluations criteria from Diagnostic and Statistical Manual for Mental Disorder—5), which will be performed by an expert neuropsychologist.

### Exclusion criteria

Exclusion criteria include presence of kidney or liver failure, or any other liver or kidney disease; presence of gastrointestinal disorders (ie, irritable bowel syndrome); presence of food intolerance; presence of heart failure, angina, pulmonary disease, cancer and cancer-related cachexia; presence of coagulation disorders; addictive or previous addictive behaviour, defined as the abuse of cannabis, opioids or other drugs, carrier of infectious diseases; presence of musculoskeletal diseases; suffering from mental illness, inability to cooperate; suffering from known cardiac conditions (eg, pacemakers, arrhythmias and cardiac conduction disturbances) or peripheral

neuropathy; regular users of any proton pump inhibitors (eg, omeprazole, lansoprazole, pantoprazole), antibiotics, anticoagulant medication or antiplatelet medications in high dose (eg, acetylsalicylic acid>200 mg per day); MMSE: results≥10 points.

## Intervention description

After receiving consent, eligible participants will enter the 2-week *run-in phase*, to stabilise the dietary protein intake at 0.9–1 g/kg ideal body weight and inform participants about the physical exercise programme. Diet quantity and quality will be assessed using a food frequency questionnaire by a dietitian, and tailored dietary advice will be provided to maintain the required calorie and protein intake. All participants will take part in the supervised physical exercise programme consisting of high-intensity aerobic endurance exercise and MST three times a week, and each session will last about 50 min. The high-intensity aerobic endurance training is carried out by walking on a treadmill with 4×4 min at 85%–95% of HRmax, interrupted by 3 min active recovery periods (~60%–70% of HRmax).[26] The MST is carried out in a seated horizontal leg press apparatus and performed as 4 sets with 4 repetitions at ~90% of maximal strength (1RM).[27] Rest periods between the sets will be 3–4 min. The training programme will take place at each centre in this multicentre study, where a room will be furnished with all the equipment needed. Physical exercise sessions will be supervised by a trained research team member and the ratio of supervisor:participants will be 1:2.

After the run-in phase, participants will be randomised into 3 intervention groups (34 participants in each group). Group A (ie, control group) will have a high-protein diet and take part in the physical exercise programme three times per week, for participants in this arm the diet plan will be adjusted to receive the same overall intake of calories and macronutrients (3 g of proteins, 4 g of carbohydrates, 11 g of fat, 4 g of fibres) that the chocolate products will provide to groups B and C. Group B will receive 30 g of 85% dark chocolate high in PP (PP≥500 mg of PP with ≥60 mg of epicatechin) in addition to high-protein diet and physical exercise. Group C will receive 30 g of 85% dark PPs functionalised chocolate with 100 mg vitamin E per day in addition to high-protein diet and physical exercise. On the days of physical exercise for the vitamin E/PP groups B and C, chocolate will be consumed in the 90 min after the end of the training. The final functionalised chocolate has been produced with a collaborating partner, Nestlè Italiana SPA (owner of 'Perugina' brand of pure chocolate products). The concentration of chocolate PP is standardised, and all the patients receive the same dose of chocolate product developed by Perugina Nestlè factory during the first phase of the project (before the initiation for the clinical trial) and thus of chocolate PP that were measured at the coordinator unit (figure 1).

## Outcomes

Primary outcome measure is the change in free-fat soft tissue mass (FFSTM, g), the change in FFSTM, g, will be assessed by means of a whole-body scan on a dual-energy X-ray absorptiometry scanner. Values at the regional level (upper limbs, lower limbs and trunk) will also be considered.

Secondary outcome measures are: (1) change in quadriceps volume and cross-sectional area (CSA) measured by using ultrasound. All ultrasound images will be acquired by an expert operator with the same ultrasound device throughout the whole study using a linear 50 mm transducer; (2) change in the torque (Nm) and rate of torque development (Nm/s) of quadriceps during maximal voluntary activation (MVC) and electrically evoked potential maximal voluntary and electrically evoked muscle contractions of the quadriceps muscle of the dominant leg will be measured using a custom-made setup; (3) change in the one repetition maximum load (kg); (4) change in the rate of force development (RFD, N/s); (5) change in submaximal and maximal oxygen consumption (mL/kg/min). Individuals will perform a 3-speed walking test, on a treadmill. First, the subjects will be asked to stand in resting condition for 2 min meanwhile the resting oxygen uptake will be recorded. Then, they will be asked to walk three 5 min bouts of walking at 80%, 100% and 120% self-selected speed, respectively. Oxygen consumption (mL/kg/min) at these three speeds will be considered for the analysis; (6) change in MMSE Score (points), the global cognitive functioning will be assessed by means of MMSE by an expert neuropsychologist; (7) change in the flow-mediated dilation (FMD, %). The brachial artery will be imaged using a high-resolution ultrasound Doppler system; (8) change in the blood flow delta peak (mL/min) during a single passive leg movement test; (9) change in the pulse wave velocity (m/s); (10) change in distance (m) during the 6 min walking test (6MWT); (11) change in time (min) during the timed up and go test (TUG); (12) change in score (number of raises) during the 30 s chair stand test; (13) changes in circadian cortisol curve (levels at four specific time throughout a day, ng/mL). Salivary cortisol will be measured using plain Sarstedt Salivette collection devices (Nürmbrecht, Germany). Immediately after sample collection, the Salivette tubes will be centrifuged for 2 min at 1000 rpm and stored at −80°C until analysis. Cortisol levels will be determined by a time-resolved fluorescence immunoassay. To assess the circadian cortisol curve the samples will be taken at 7:00, 11:00, 15:00 and 20:00; (14) acute cortisol response to the exercise (delta percentage between before and after a training session, %); (15) change in interleukin 6 (pg/mL) and IGF-1 (ng/mL) concentrations; (16) change in malondialdehyde (µM); (17) change in mRNA expression, RNA samples will be processed by following the specific platform protocols and the final results will be bioinformatically analysed. Expression analysis software and pipelines will be used to analyse the differential expression profiles of the

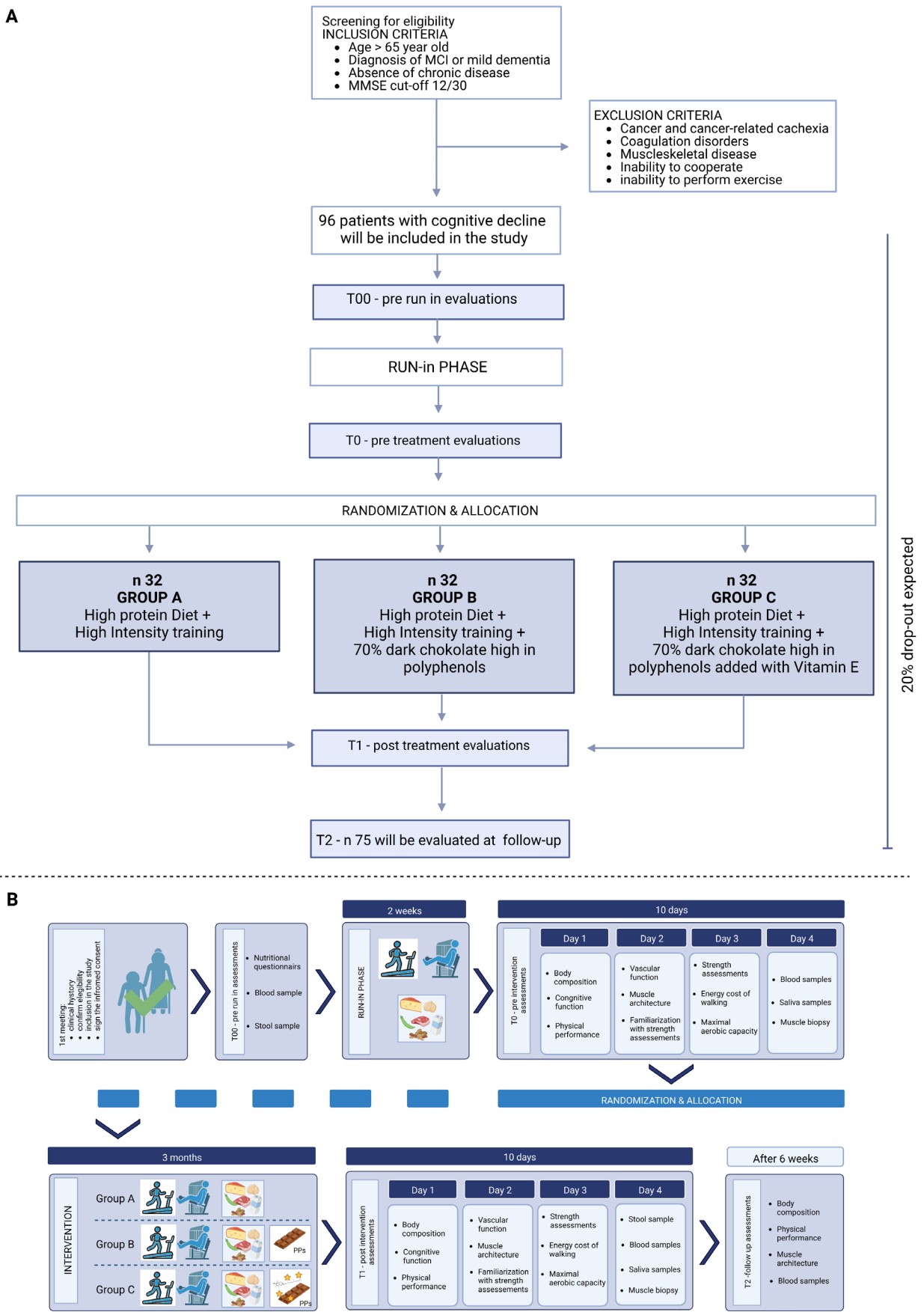

**Figure 1** Study design and Individual's involvement. Flow chart of the study design showing information on enrolment, group assignment, follow-up and data analysis A) together with the flow chart of individual's involvement (B). MMSE, Mini–Mental State Examination; MCI, Mild Cognitive Impairement.

selected genes. We will also analyse SUB-NETWORKS, this is to see the relationships that exist between the different transcripts to try and find common molecular pathways; (18) change in the microbiota composition: Bacterial DNA will be extracted from faecal samples and then amplified and sequenced using a high-throughput next-generation sequencing platform able to generate million short sequences (reads) per single run. Sequences will be then processed using a bioinformatic pipeline whose steps that can be summarised as follows: raw data collection, data cleaning, assembly, gene prediction, taxonomic annotation, gene and protein abundance estimation; (19) change in the muscle histology and fibre typing; (20) change in the muscle mitochondrial respiration; (21) change in the muscle in vitro force characteristics. After the biopsy, fibre bundles of 4–6 mm in length and 0.5 mm in diameter will be dissected from the samples and immersed in skinning solution to which the non-ionic detergent Brij 58 had been added; (23) change in the muscle single fibre measurements; (24) change in the muscle cytokine mRNA; (25) change in the muscle redox status; (26) change in the muscle proteomics.

A portion of the frozen muscle will be thawed on ice and prepared for proteomic analysis as previously described. A global label-free proteomic approach will be used using an Ultimate 3000 RSLC nano system coupled to a QExactive mass spectrometer. Data analysis will be performed using Proteome Discover and Peaks V.7.

### Clinical and physiological assessments
All assessments are presented in schematic figure 2 for everyone.

### Body composition
Dual X-ray absorptiometry will be used for segmental and total body muscle mass and fat mass and bone mineral density.[28] The quadriceps volume and CSA will be measured by the same ultrasound device throughout the whole study.[29]

### Physical function tests
6MWT, TUG, and 30 s chair stand test. For measuring the strength component, individuals will undergo MVC and electrically evoked potential, to estimate the role of central fatigue; RFD and peak force will be assessed to provide data on the relative neural and muscular contribution, respectively, to the muscle force development. To assess the energy cost of walking and maximal oxygen consumption. Individuals will perform a 3-speed walking test on a treadmill.

### Cognitive assessment
Patients will undergo a full neuropsychological battery of tests to explore verbal, visuospatial memory, working memory, attention, dual task and executive functions. The global cognitive functioning will be assessed with MMSE. The Neuropsychiatric Inventory will be used to evaluate the presence, frequency and severity of behavioural disorders. The Geriatric Depression Scale will be used to assess the presence and the severity of depression.

### Vascular function
Vascular function will be assessed by means of FMD and single passive leg movement.[30]

### Saliva cortisol
Cortisol will be collected using Salivette with a cotton swab (Sarstedt, AG & Co. KG, Nümbrecht, Germany) at the beginning and at the end of the 3-month treatment. Swab samples will be collected: (1) to measure acute response to a single exercise training session, saliva samples will be collected before starting the sessions and right at the end of one of it; (2) to measure the circadian curve, participants will be asked to collect saliva samples in a regular day (ie, without exercise training) at four specific time points (7:00, 11:00, 15: and 19:00).[31 32]

### Blood assays
Whole blood will be examined for glycosylation and transcriptomics. Plasma samples will be obtained for protein oxidation, lipid peroxidation, general hormones, ELISA kit assays. Furthermore, the analysis will include gas chromatography–mass spectrometry (GC-MS) and liquid chromatography (LC)-MS/MS determinations of plasma levels of chocolate PPs and vitamin E metabolites.[33] Serum sample will be analysed for untargeted metabolomics, this latter is expected to provide a generic fingerprint of the response to the proposed nutritional intervention and to training activities. Peripheral blood mononuclear cells (PBMCs) will be used to extract proteins and mRNA for the analysis of genes involved in vitamin E metabolism (such as CYP4F2).

### RNA extraction
Total RNA containing miRNA will be isolated by the mirVana miRNA Isolation Kit (Ambion, Austin, Texas, USA) according to the manufacturer's directions. RNA concentration and purity will be assessed with 260/280 ratio using a Genequant Pro Classic spectrophotometer (GE Healthcare, Spain) and RNA integrity by capillary electrophoresis using the RNA 6000 Nano Lab-on-a-Chip kit and the Bioanalyzer 2100 (Agilent Technologies, Santa Clara, California, USA).

### Gene expression profiling
mRNA profiling will be performed using Clariom D-Human Array (Thermo Fisher Scientific). This array comprises >542 000 transcripts constituting over 134 000 gene-level probe sets. Microarray experiments were conducted according to the manufacturer's instructions (Thermo Fisher Scientific). Briefly, 150 ng total RNA will be labelled using GeneChip WT Plus Reagent Kit. The labelling reaction will be hybridised on the array using Hybridization Oven 645 at 45°C for 16 hours. The arrays are stained with Fluidics Station 450 using fluidics script FS450_0001 and then scanned on a GeneChip Scanner 3000 7G. GeneChip Command Console software supplied

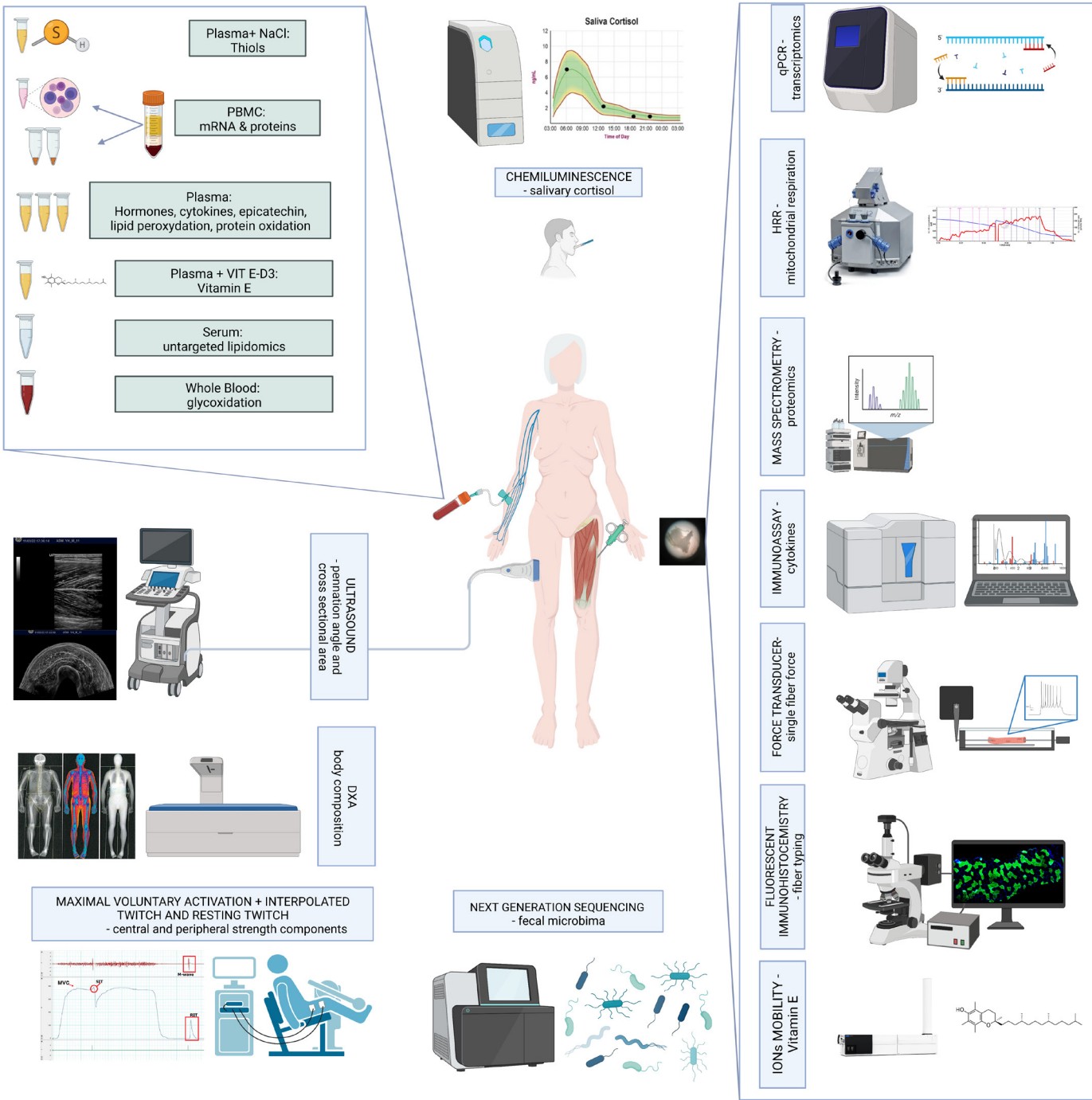

**Figure 2** Methods of primary, secondary and explorative outcomes. Primary outcomes include: body composition, specifically free-fat mass in the lower limbs, measured by means of DXA. Another primary outcome consists of maximal voluntary activation measured by MVS+IT+RT technique. Secondary outcomes include cross-sectional area of the quadriceps muscle and pinnation angle, salivary cortisol, measured by means of chemiluminescence and biological outcomes such as: thiols, mRNA and proteomics, vitamin E, untargeted lipidomic, glycoxidation, hormones, cytokines, epicatechin, lipid peroxidation and protein oxidation. Furthermore, secondary outcomes include mitochondrial respiration measured on muscle fibbers, single fibre force, fibber typing, cytokines, transcriptomics and proteomic measured in the muscle fibbers. Exploratory outcomes include vitamin E measured in muscle fibber and faecal microbiome. DXA, dual X-ray absorptiometry. PBMC, periferal blood mononuclear cells; HRR, high resolution resirometry.

by Thermo Fisher Scientific will be used to perform gene expression analysis.

## Data analysis of microarrays

Data (.CEL files) are analysed and filtered using the Partek Genomic Suite V.6.6 software (Partek, St. Louis, Missouri, USA). Input files are normalised with the RMA algorithm for gene array on core meta probe sets or miRNAs. A one-way analysis of variance (ANOVA) will be performed with the Partek Genomics Suite across all samples. Statistically significant small non-coding RNAs and mRNAs between the different groups studied will be identified by using a model ANOVA of $p < 0.05$. The imported data will be analysed by principal component analysis to determine the significant sources of variability in the data. Differentially expressed genes will be imported into Pathway Studio.

## Muscle biopsy

A sample of skeletal muscle will be obtained with a 13-gauge tru-cut needle from the vastus lateralis muscle. This will be used for the following assays: (1) Assessment of muscle cytokine mRNA for a panel of proinflammatory cytokines examined by quantitative real-time PCR (qPCR). RNA will be isolated using the standard Trizol extraction method and purified using RNeasy clean-up kit; cDNA will be synthesised using iScript first strand kit from 1 µg of isolated RNA. (2) Muscle redox status will be undertaken by analysis of the reduced and oxidised glutathione contents of the biopsy together with analysis of the redox status of mitochondria through analysis of the proportion of peroxiredoxin three in the oxidised form, and cytosol by analysis of the proportion of peroxiredoxin two in the oxidised form[34] (3) Muscle proteomics: a global label-free proteomic approach will be used at Ultimate 3000 RSLC nano system coupled to a QExactive mass spectrometer. Data analysis will be performed using Proteome Discover and Peaks V.7 software.[34] (4) Determination of mRNA expression: RNA samples will be processed by following the specific platform protocols and the results will be bioinformatically analysed. Expression analysis software and pipelines will be used to analyse the differential expression profiles of the selected genes. We will also analyse SUB-NETWORKS; this is to see the relationships that exist between the different transcripts to try and find common molecular pathways. (5) Mitochondrial respiration derived from the maximal respiratory capacity and in vitro force generation of types 1 and 2 fibres assessed on muscle fibres obtained from the muscle biopsy.[35] (6) Targeted metabolomics of chocolate PP and vitamin E by GC-MS and LC.

## Provision of metabolomics

Main analyses will include GC-MS and LC-MS/MS determinations of blood levels of chocolate PP and vitamin E metabolites, as well as untargeted metabolomics of serum lipids, this latter is expected to provide a generic fingerprint of the response to the proposed nutritional intervention and to HIT. Vitamin E (as alpha-tocopherol) and its long-chain metabolites deriving from CYP450 and free radical dependent biotransformation will be investigated for the first time in human muscle tissue. Moreover, the modification of their levels in this tissue in response to vitamin E supplementation (provided by the innovative food product) is a novel investigation. Bioinformatics analysis will be carried out with the Lipostar software and will provide a model of interpretation of the molecular response and mechanistic effects of the nutritional and physical activity intervention. The model will include the identification of pathways and molecular interactions, as well as of candidate biomarkers of individual and target population response to the proposed intervention.

## Sample size

Considering an alpha=0.05, a power=0.8 and the 20% of estimated drop out, we aim to recruit 102 subjects (34 in each group). The main outcome is 'muscle mass', and for all the groups treatment duration will be 6 months. In 6 months in the target population, the loss of muscle mass is assumed to be 1.0%–1.5% (±0.5%).[36] In control group (group A), which includes people undergoing targeted exercise, the expected increase is 2%–1.5% (±0.5%). In treatment groups (groups B and C), the median average expected increase at second follow-up is 4%–2% (±0.5%) and 1.5%–4% (±0.5%) at 6 months.[37] The rate of lost at follow-up, derived from previous studies, is 20% (±2%). The correlation between repeated measures is assumed to be 0.5, variance explained by the between-subjects effect 6.25 and error variance 65. All estimates were performed using Stata V.16.1 (StataCorp LP, College Station, Texas, USA) by 'power repeated' command.

## Allocation and blinding

This is a randomised, double-blinded, controlled trial. A blocked randomisation approach will be applied to reach the required number for each group. A statistician will create a randomisation list including the groups (A, B or C) allocated to each subject, the blocks (after specifying the block size) will be randomly chosen to determine the assignment of all participants. Neither the participants nor research team will know the assignments of the intervention groups including type of chocolate (vitamin E and poly phenols, or poly phenols). Also, the participant in the control group (A) will not be aware of the chocolate consumption available for groups B and C.

## Patient and public involvement

There were no implications for the patient and public involvement.

## Statistical analysis

Statistical analysis will be conducted under the supervision of an expert in biostatistics and with the support of LIPOSTAR software. The primary data analysis will be two-way repeated measures ANOVA, including age and gender as covariates, with 'time' as the within-group factor

and 'treatment' as the between-groups factor to calculate difference between groups. In the presence of significant effects, a multiple comparisons tests with Bonferroni's correction will be performed.

We will also employ mixed model analysis to account for both fixed effects, and random effects, to capture variability or correlations within the data that are not under our direct control. Also, since we are dealing with multiple sources of variation, such as repeated measurements on the same subjects, nested or hierarchical data structures and potentially correlated factors. Mixed model analysis can provide a more accurate and robust understanding of the relationships within our data.

The familywise alpha level for significance will be set at 0.05 (two tails), with Bonferroni's correction when needed, for all the analyses. This study is inherently generating extensive clinical and omics data which will be analysed using robust and appropriate data analysis and with machine learning algorithms.

## Data monitoring
A log–diary will be kept by each participant and will be checked weekly by the investigators and collaborators. In the diary, participants will include information about possible adverse events caused by assessment procedures or related to the diet and training, any important points about the response to the interventions, any possible discomfort experienced during or after the training, or notes regarding diet and supplementation. Professor Gianluca-Svegliati Baroni of the Gastroenterology Division of the University Hospital of Ancona, Italy, will serve as external scientific supervisor of the clinical trial. He is an expert in clinical and preclinical studies of human nutrition and metabolism. The supervisor will advise on specific code: CHOKO-AGE Data: 10/06/2021 Version:1 30 tasks and monitor the different phases of clinical trial from organisation to implementation of activities, data gathering and evaluation/interpretation. The quality assurance standards of University of Verona will be adopted to monitor the clinical trial. A delegate of this University will be nominated to perform the monitoring of the different phases of the trial using internal Standard Operating Procedures. The entire set of clinical procedures, operator's activity and collection of experimental data will be verified during a series of visits by the monitor that will occur at the beginning and the end of each time point in the study (time T00 to T3).

## Limitation of the study
There are some limitations to this study, the RCTs population are mostly white European descendant that are not representative of all older population. This study will not be able to account for effect of intervention and change beyond the study duration, making it challenging to evaluate the long-term sustainability of the intervention. By nature, it is not possible to blind a participant for exercise intervention, and this may have an effect on the participant's lifestyle choices beyond what can be captured with the study.

## Ethics and disseminations
This protocol and the informed consent forms have been reviewed and approved by the applicable ethical committees of the *Azienda Ospedaliera Universitaria Integrata* of Verona with respect to scientific content and compliance with applicable research and human subjects' regulation. Thereby, this protocol is ethically conducted following the last revision of the Declaration of Helsinki as well as the Declaration of Oviedo. The protocol of the study is designed and will be conducted in order to adhere to good clinical practice principles and procedures and in compliance with the Italian legislation, as described in the following documents and accepted by the investigators of the study with their signature: (1) ICH Harmonized Tripartite Guidelines for Good Clinical Practice 1996; (2) Directive 91/507/EEC, The Rules Governing Medicinal Products in the European Community; (3) D. L.vo n.211 of 24 June 2003; (4) D. L.vo n.200 6 November 2007; (5) D.M. 21 December 2007. NIH Clinical trial identification ID and number: JPI- ERAHDL NCT05343611.

## Consent, confidentiality, declaration of interests
A member of the research team will introduce the trial to individuals who show an interest in taking part in the study and their caregivers. Individuals will also receive information sheets. A member of the research team will discuss the trial with potential participants and their caregivers considering the information provided in the information sheets. Individuals and caregivers will then be able to have an informed discussion with the member of the research team who will obtain written informed consent from the persons willing to participate in the trial. All study-related information will be stored securely at the study site. All participant information will be stored in locked file cabinets in area with limited access. All laboratory specimens, reports, data collection, process and administrative forms will be identified by a coded ID number only to maintain participant confidentiality. All records that contain names or other personnel identifier will be stored separately from study records identified by code number. All local databases will be secured with password-protected access systems. Participants information will be stored for 10 years. Forms, lists, logbook, appointment books and any other listing that link to participant ID numbers to other identifying information will be stored in a separate, locked file in an area with limited access.

## Conflict of interest
None declared.

## Data access
Access to the database to start a data entry session is protected by a username and password. The following precautionary measures are taken to ensure data privacy and to prevent data manipulation and loss: (1) access to data is restricted to authorised members only. The

authorised members are the researchers directly involved in the study who will collect, analyse and interpret the data; (2) the network is protected by a firewall; (3) internet connection is encrypted with a digital certificate (Secure Sockets Layer technology); (4) the database is on a server, password protected, which is changed periodically; (5) access to the database is password protected and is accessible only to those responsible for the centre; (6) periodic backups are performed; (7) and places of conservation (eg, the paper materials related to clinical evaluations will be stored in cabinets, whose keys will be in possession only of the persons authorised by the persons in charge of the study). Personal data of the subjects will be exclusively accessible for investigators and collaborators/coinvestigators, monitors and auditors of the promoter, and/or for the competent authority, in agreement with what is included in the informed consent. The coordinator and the investigators will inform the patients in a clear and thorough way about the modalities of personal data treatment before their participation to the experiment, as established by the applicable privacy laws.

## Dissemination policy

A consortium agreement has been established for the dissemination policy. Each unit is encouraged to publish in written form, oral presentation or making public in any other form (including electronic publication on the internet), hereinafter referred to as the 'publication', the results of the project, including but not limited to solely generated foreground, after providing a copy of the draft publication to the other parties at least 30 calendar days prior to submission for publication or disclosure. The other parties may, by giving written notice to the unit requesting the publication (a 'confidentiality notice'), require the publishing unit to: (1) amend the publication in order to remove any of its background that is confidential information for the duration of the confidentiality obligation, or to remove any of its foreground or (2) amend the publication if it can reasonably demonstrate that the manuscript compromises disproportionally its academic or legitimate interests. The other unit shall give such confidentiality notice within 30 calendar days after receipt of the draft publication. The confidentiality notice shall contain specific details to be amended and the party-seeking disclosure shall delete said specific details in as far as they comprise the other units's confidential information from the publication or amend said specific details as not to consortium agreement 'Choko-Age project' disproportionally harm the requesting unit's academic or legitimate interests. If no objection is made within the time limit stated above, the publication is permitted.

If an objection has been raised, the involved units shall discuss how to overcome the justified grounds for the objection on a timely basis (eg, by amendment to the planned publication before publication). The objecting unit shall not unreasonably continue the opposition if appropriate actions are performed following the discussion. In any event, any delays of a publication shall not exceed a period of more than 90 calendar days from the date of the notification from the unit-seeking disclosure. Contributors to the results obtained through work on the project published shall be named and accredited in the publication in accordance with academic custom. Other aspects concerning result dissemination and the communications strategy of the project are presented in the 'communication and dissemination plan' enclosed as Annex IV to the present agreement.

**Author affiliations**
[1]Cellular, Computational and Integrative Biology, University of Trento, Trento, Italy
[2]Centre for Integrated Research into Musculoskeletal Ageing (CIMA), Musculoskeletal & Ageing Science, University of Liverpool, Liverpool, UK
[3]Department of Medicine and Surgery, University of Perugia, Perugia, Italy
[4]Department of Medicine and Surgery, Bioscience and Medical Embryology Division, University of Perugia, Perugia, Italy
[5]Department of Physiology, Faculty of Medicine, University of Valencia, Valencia, Spain
[6]Department of Neuroscience, Biomedicine and Movement (DNBM), University of Verona, Verona, Italy
[7]Department of Pharmaceutical Sciences, Lipidomics and Micronutrient, University of Perugia, Perugia, Italy
[8]Freshage Research Group, Department of Physiology, Faculty of Medicine, Centro de Investigación Biomédica en Red Fragilidad y Envejecimiento Saludable-Instituto de Salud Carlos III (CIBERFES-ISCIII), University of Valencia, Valencia, Spain
[9]Department of Health and Social Sciences, Molde University College, Molde, Norway
[10]Molecular Horizon srl, I-06084, Bettona, Italy
[11]St Olavs Hospital Trondheim University Hospital, Trondheim, Norway
[12]Department of Pharmaceutical Sciences, University of Perugia, Perugia, Italy

**Contributors** AP and MI were the co-lead and wrote the protocol according to the project descriptive, created figures and structured the manuscripts. FG was the PI of the project. All authors contributed to the conception and design of the work, data acquisition, drafting and revising the study and are accountable for its intellectual content. This is the first version of the protocol submitted to *BMJ Open*.

**Funding** To intervene with a high-calorie micronutrient-rich diet in elderly patients at risk of PEM could be problematic for different reasons including the difficulty to find palatable food items suitable to satisfy these nutritional criteria. The need for specific innovation in this area of food industry and human nutrition has recently been addressed in the call PREVNUT promoted by the 'Joined Programming Initiatives—Healthy Diet for a Healthy Life' (JPI-HDHL) supported by the EU program Horizon 2020 under the umbrella of ERA-Net/ERA-HDHL and the international network of funding organizations reported in (https://www.healthydietforhealthylife.eu/index.php/call-activities/calls/98-calls-site-restyling/583-era-hdhl-2020). This project has received funding from MUR, Ministero dell'Università e della Ricerca (DD17080-24/11/2021) (Italy); Instituto de Salud Carlos III (ISCIII) (Ref. AC20/00026) (Spain); RCN, The Research Council of Norway (Norway) and UKRI, Biotechnology and Biological Sciences Research Council (BBSRC) (grant number BB/V019821/1) (UK), under the umbrella of the European JPI HDHL and of the ERA-NET Cofund ERA-HDHL (GA N° 696295 of the EU Horizon 2020 Research and Innovation Programme).

**Competing interests** None declared.

**Patient and public involvement** Patients and/or the public were not involved in the design, or conduct, or reporting, or dissemination plans of this research.

**Patient consent for publication** Consent obtained directly from patient(s).

**Provenance and peer review** Not commissioned; externally peer reviewed.

**ORCID iDs**
Masoud Isanejad http://orcid.org/0000-0002-3720-5152
Sara Tortorella http://orcid.org/0000-0001-9691-8323

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
