## [Reviewer comments · BMJ Open]

ARTICLE DETAILS

TITLE (PROVISIONAL)	Randomized Controlled Trial Combining vitamin E-functionalized Chocolate with physical exercise to reduce the risk of protein-energy malnutrition in pre-dementia Aged people: Study protocol for Choko-Age
AUTHORS	Pedrinolla, Anna; Isanejad, Masoud; Antognelli, Cinzia; Bartolini, Desirée; Borrás, Consuelo; Cavedon, Valentina; Di Sante, Gabriele; Migni, Anna; Mas-Bargues, Cristina; Milanese, Chiara; Baschiroto, Claudia; Modena, Roberto; Pistilli, Alessandra; Rende, Mario; Schena, Federico; Stabile, Anna Maria; Vincenzo Telesa, Nicola; Tortorella, Sara; Hemmings, Kay; Vina, Jose; Wang, Eivind; McArdle, Anne; Jackson, Malcolm; Venturelli, Massimo; Galli, Francesco

VERSION 1 – REVIEW

REVIEWER	Yassine, Hussein University of Southern California
REVIEW RETURNED	26-Feb-2023

GENERAL COMMENTS	The protocol paper is missing key aspects: The rationale why three groups assigned to high protein diet with and without chocolate polyphenol or exercise intervention is going to increase muscle mass in this population is not very clear. Why the need for three different groups? On the surface, the study appears underpowered with such small sample size. Inclusion and exclusion criteria appear incomplete. How do they define MCI? How do they exclude dementia? This MMSE cut off of 12 would include individuals with dementia. In the exclusion criteria, what about dementia medications, TBI, psychizophrenia? This looks vague. What about muscle mass at baseline? Any considerations for individuals with sarcopenia at baseline? The sample size determination is inadequate. What would they expect the effect size to be and based on what? Data analysis section is inadequate. What models are they using to analyze the primary, secondary, and exploratory variables? They describe a two way ANOVA but would they not need a mixed model analysis plan to account for repeated analysis? How would they analyze the omics data? Are they allowed to do a such a trial without a DSMB? They mention an external advisor but is that sufficient? Who appoints this advisor and how the reporting is planned are not described.
--

	The limitations of the trial methodology and interpretation of findings are not addressed. Figure 2 can be improved by specifying which one is the primary or secondary or exploratory outcome.
--	---

REVIEWER	Rodriguez-Mateos, Ana King's College London
REVIEW RETURNED	14-May-2023

GENERAL COMMENTS	Overall, this is a well written study protocol and a very interesting study. I have few comments for the authors to consider. A general comment is to double check that the information presented here is the same to the one available and registered in clinicaltrials.gov. Best to use the same language to avoid confusion. I saw for example that in there you have the study described as single blinded (participants) instead of double blinded. Please make sure that the primary & secondary outcomes listed online are exactly the same as presented here (ie you have 24 secondary outcomes listed online, I dont see some like Pulse Wave Velocity or gut microbiome being mentioned here) 1) Eligibility criteria - assume men and women are included ? 2) Intervention description - It would be useful to know exactly how many visits the study consists of, and how many weeks the intervention will last. Some important details, like when the neuropsychological test battery, or the vascular function measurements will be collected, would be useful to include here (or in another section). In terms of the polyphenols contained in the chocolate, why it says " higher or equal" to 500 mg? shouldn't be exactly the same for all chocolates (ie 500 mg PP per 30g of chocolate) ? Is this going to be standardised throughout the study, as it is quite important? 3) Outcomes: Please clearly specify which one is the primary outcome, fat free soft tissue mass? or change of lower limb muscle mass? You have fat free soft tissue mass in clinicaltrials.gov. Also be more specific regarding secondary outcomes ie which aspects of cognitive function ? A full list of secondary/tertiary/exploratory outcomes will be useful, specially as not all the ones described in the clinical and physiological assessment section are mentioned here (and clinical trials.gov) 4) Clinical and physiological assessments - it would be useful to add a bit more info on some of the outcomes such as which neuropsychological battery will be used, or how the FMD will be measured, for example. Also, only in some cases it is reported when the measurements will be assessed (ie:"saliva will be collected at the beginning and at the end of the 3 months treatment "....), this information should be added for all outcomes as this is very important information. 5) Provision of metabolomics - which plasma will be used, before and after 3 months of treatment? which targeted analysis will be conducted ? 6)Allocation and blinding - which parameters will be used for the blocked randomization? Minor comments: Page 4 line 9 - Verb missing "European older adults ARE at high risk, ..."
--

	Page 4 Lines 19 & 20 - there is something missing in the sentence "leading to a disruption in the production of glucose, AND? reduction in energy supply to the muscles" and in the sentence starting with "The, and.." Page 5 line 57 - typo in "trials" (RCTs) should be trials
--	---

VERSION 1 – AUTHOR RESPONSE

Reviewer: 1

The protocol paper is missing key aspects: The rationale why three groups assigned to high protein diet with and without chocolate polyphenol or exercise intervention is going to increase muscle mass in this population is not very clear. Why the need for three different groups? On the surface, the study appears underpowered with such small sample size.

Revised text: The main hypothesis of this study is whether the antioxidant and cytoprotective functions of vitamin E, combined with the cortisol-lowering effect of chocolate polyphenols, may help prevent the age-dependent decline of mitochondrial function and nutrient metabolism in skeletal muscle. We test this hypothesis in a group of older people with stabilized dietary protein intake and physical activity levels. This is important as we are able to account for these two major lifestyle factors and age-dependent muscle function decline before testing the metabolic effects of chocolate polyphenols and vitamin E.

The power and sample size calculation is explained in page 11 of this manuscript- we have revised the text now:

Considering an alpha = 0.05, a power = 0.8, and the 20% of estimated dropout, we aim to recruit 102 subjects (34 in each group). The main outcome is "muscle mass", and for all the groups, the treatment duration will be 6 months. In 6 months in the target population, the loss of muscle mass is assumed to be 1.0-1.5% (+/- 0.5%) (38). In the Control group (Group A), which includes people undergoing targeted exercise, the expected increase is 2% 1.5% (+-0.5%). In treatment groups (Groups B and C) the median average expected increase at second follow-up is 4% 2% (+-0.5%), and 1.5% 4% (+-0.5%) at 6 months (39). The rate of loss at follow-up, derived from previous studies, is 20% (+-2%). The correlation between repeated measures is assumed to be 0.5, with variance explained by the between-subjects effect of 6.25 and error variance 65. All estimates were performed using Stata v.16.1 (StataCorp LP, College Station, TX, USA) by the "power repeated" command.

Inclusion and exclusion criteria appear incomplete. How do they define MCI? How do they exclude dementia? This MMSE cut off of 12 would include individuals with dementia. In the exclusion criteria, what about dementia medications, TBI, schizophrenia? This looks vague. What about muscle mass at baseline? Any considerations for individuals with sarcopenia at baseline?

We have now revised the manuscript according to the clinical trial registration:

Inclusion Criteria:

Presence of Mild Cognitive Impairment or Mild Dementia. Recruited individuals will be assessed by means of Neuropsychological tests (Mini-Mental State Examination, evaluations criteria from Diagnostic and Statistical Manual for Mental Disorder-5) which will be performed by an expert Neuropsychologist.

Exclusion Criteria:

Presence of kidney or liver failure, or any other liver or kidney disease; Presence of gastrointestinal disorders (i.e. irritable bowel syndrome); Presence of food intolerance; Presence of heart failure, angina, pulmonary disease, cancer, and cancer-related cachexia; Presence of coagulation disorders; Addictive or previous addictive behaviour, defined as the abuse of cannabis, opioids or other drugs, carrier of infectious diseases; Presence of musculoskeletal diseases; Suffering from mental illness, inability to cooperate; Suffering from known cardiac conditions (e.g. pacemakers, arrhythmias, and cardiac conduction disturbances) or peripheral neuropathy; Regular users of any proton pump inhibitors (e.g., omeprazole, lansoprazole, pantoprazole), antibiotics, anticoagulant medication or antiplatelet medications in high dose (es: acetylsalicylic acid >200mg x day); Mini Mental State (MMSE): results \geq 10 points.

The sample size determination is inadequate. What would they expect the effect size to be and based on what?

Considering an $\alpha=0.05$, a power = 0.8, and the 20% of estimated dropout, we aim to recruit 102 subjects (34 in each group). The main outcome is “muscle mass”, and for all the groups, the treatment duration will be 6 months. In 6 months in the target population, the loss of muscle mass is assumed to be 1.0-1.5% (+/- 0.5%) (38). In the Control group (Group A), which includes people undergoing targeted exercise, the expected increase is 2% 1.5% (+-0.5%). In treatment groups (Groups B and C) the median average expected increase at second follow-up is 4% 2% (+-0.5%), and 1.5% 4% (+-0.5%) at 6 months (39). The rate of loss at follow-up, derived from previous studies, is 20% (+-2%). The correlation between repeated measures is assumed to be 0.5, with variance explained by the between-subjects effect of 6.25 and error variance 65. All estimates were performed using Stata v.16.1 (StataCorp LP, College Station, TX, USA) by the “power repeated” command.

Data analysis section is inadequate. What models are they using to analyse the primary, secondary, and exploratory variables? They describe a two-way ANOVA but would they not need a mixed model analysis plan to account for repeated analysis?

We appreciate this comment. As stated in the text the data generated by this study will be analysed with appropriate method and in consultation with biostatistician partners of the project, as such is the mixed model analysis for repeated measurements. A two-way ANOVA will be carried out for the primary outcome and to report change in fat-free mass.

How would they analyse the omics data?

In the revised text we included further information in page 11:

Main analyses will include GC-MS and LC-MS/MS determinations of blood levels of chocolate polyphenols, and vitamin E metabolites, as well as untargeted metabolomics of serum lipids, this latter

is expected to provide a generic fingerprint of the response to the proposed nutritional intervention and HIT. Vitamin E (as alpha-tocopherol) and its long-chain metabolites deriving from CYP450 and free radical-dependent biotransformation will be investigated for the first time in human muscle tissue. Moreover, the modification of their levels in this tissue in response to vitamin E supplementation (provided by the innovative food product) is a novel investigation. Bioinformatics analysis will be carried out with the Lipostar software and will provide a model of interpretation of the molecular response and mechanistic effects of the nutritional and physical activity intervention. The model will include the identification of pathways and molecular interactions, as well as of candidate biomarkers of individual and target population response to the proposed intervention

Are they allowed to do a such a trial without a DSMB? They mention an external advisor but is that sufficient? Who appoints this advisor and how the reporting is planned are not described.

There was no DSMB allocated for this project, we have added further details regarding the data monitoring strategy to page 12 and 13:

A log diary will be kept by each participant and will be checked weekly by the investigators and collaborators. In the diary, participants will include information about possible adverse events caused by assessment procedures or related to the diet and training, any important points about the response to the interventions, any possible discomfort experienced during or after the training, or notes regarding diet and supplementation. Prof. Gianluca-Svegliati Baroni of the Gastroenterology Division of the University Hospital of Ancona, Italy will serve as external scientific supervisor of the clinical trial. He is an expert in clinical and preclinical studies of human nutrition and metabolism. He will advise on specific Code: CHOKO-AGE Data: 10/06/2021 Version:1 30 tasks and monitor the different phases of the clinical trial from organization to implementation of activities, data gathering, and evaluation/interpretation. The quality assurance standards of the University of Verona will be adopted to monitor the clinical trial. A delegate of this University will be nominated to perform the monitoring of the different phases of the trial utilizing internal SOPs. The entire set of clinical procedures, operator's activity, and collection of experimental data will be verified during a series of visits by the monitor that will occur at the beginning and the end of each time point in the study (Time T00 to T3).

The limitations of the trial methodology and interpretation of findings are not addressed.

we added not to page 14 a section for limitation: There are some limitations to this study, the RCTs population are mostly white European descendants that are not representative of all older population. This study will not be able to account for the effect of the intervention and change beyond the study duration, making it challenging to evaluate the long-term sustainability of the intervention. By nature, it is not possible to blind participants for exercise intervention and this may have an effect on the participant's lifestyle choices beyond what can be captured with the study.

Figure 2 can be improved by specifying which one is the primary or secondary or exploratory outcome.

We have now added text box to figure 2. Figure 2 includes all the methods used to collect primary, secondary, and exploratory outcomes. Primary outcomes include body composition, specifically free-fat mass in the lower limbs, measured by means of DXA. Another primary outcome consists of maximal voluntary activation measured by the MVS+IT+RT technique. Secondary outcomes include a cross-

sectional area of the quadriceps muscle and pinnation angle, salivary cortisol, measured by means of chemiluminescence, and biological outcomes such as thiols, mRNA and proteomics, vitamin-E, untargeted lipidomic, glycoxidation, hormones, cytokines, epicatechin, lipid peroxidation, and protein oxidation. Furthermore, secondary outcomes include mitochondrial respiration measured on muscle fibers, single fibre force, fiber typing, cytokines, transcriptomics, and proteomics measured in the muscle fibres. Exploratory outcomes include vitamin-e measured in muscle fibres, and faecal microbiome.

Reviewer: 2

Dr. Ana Rodriguez-Mateos, King's College London

Comments to the Author: Overall, this is a well written study protocol and a very interesting study. I have few comments for the authors to consider.

A general comment is to double check that the information presented here is the same to the one available and registered in clinicaltrials.gov, Best to use the same language to avoid confusion. I saw for example that in there you have the study described as single blinded (participants) instead of double blinded.

We have now updated both the manuscript text and clinical trial registration to reflected this and other discrepancies.

Please make sure that the primary & secondary outcomes listed online are exactly the same as presented here (i.e. you have 24 secondary outcomes listed online, I don't see some like Pulse Wave Velocity or gut microbiome being mentioned here).

We have updated both documents and added the following text to page 7 and 8:

Outcomes

Primary Outcome Measure: change in free-fat soft tissue mass (g) Change in free-fat soft tissue mass, (FFSTM, g), will be assessed by means of a whole-body scan on a dual-energy X-ray absorptiometry scanner. Values at the regional level (upper limbs, lower limbs, and trunk) will also be considered.

Secondary Outcome Measure: 1) Change in Quadriceps volume and cross-sectional area measured by using ultrasound. All ultrasound images will be acquired by an expert operator with the same ultrasound device throughout the whole study using a linear 50 mm transducer. 2) Change in the torque (Nm) and rate of torque development (Nm/s) of quadriceps during Maximal Voluntary Activation and electrically evoked potential Maximal voluntary and electrically evoked muscle contractions of the quadriceps muscle of the dominant leg will be measured utilizing custom-made setup. 3) Change in the one repetition maximum load (kg). 4) Change in the Rate of Force Development (N/s). 5) Change in submaximal and maximal oxygen consumption (ml/kg/min) Individuals will perform a 3-speed walking test, on a treadmill. First, the subjects will be asked to stand in resting condition for 2 minutes meanwhile the resting oxygen uptake will be recorded. Then they will be asked to walk three 5- minute bouts of

walking at 80%, 100%, and 120% self-selected speed respectively. Oxygen consumption (ml/kg/min) at these three speeds will be considered for the analysis. 6) Change in Mini-Mental State Examination score (points), the global cognitive functioning will be assessed by means of Mini-Mental State Examination by an expert Neuropsychologist. 7) Change in the Flow-mediated dilation (%) The brachial artery will be imaged using a high-resolution ultrasound Doppler system. 8) Change in the Blood flow delta peak (ml/min) during a Single Passive-Leg Movement test. 9) Change in the Pulse Wave Velocity (m/s). 10) Change in distance (meters) during the 6-minute walking test. 11) Change in time (min) during the Time-up and go test (TUG). 12) Change in the score (number of raises) during the 30 seconds Chair-stand test. 13) Changes in the Circadian Cortisol curve (levels at 4 specific times throughout a day, ng/mL) Salivary cortisol will be measured using plain Sarstedt Salivette collection devices (Nürmbrecht, Germany). Immediately after sample collection, the Salivette tubes will be centrifuged for 2 minutes at 1000 rpm and stored at -80C until analysis. Cortisol levels will be determined by a time-resolved fluorescence immunoassay. To assess the circadian cortisol curve the samples will be taken at 7 am, 11 am, 3 pm, and 8 pm. 14) Acute Cortisol response to the exercise (delta percentage between before and after a training session, %). 15) Change in IL-6 (pg/mL) and IGF-1 (ng/mL) concentrations. 16) Change in malondialdehyde (MDA, μ M). 17) Change in mRNA expression, RNA samples will be processed by following the specific platform protocols, and the final results will be bioinformatically analysed. Expression analysis software and pipelines will be used to analyse the differential expression profiles of the selected genes. We will also analyse SUB-NETWORKS, this is to see the relationships that exist between the different transcripts to try and find common molecular pathways. 18) Change in the microbiota composition: Bacterial DNA will be extracted from faecal samples and then amplified and sequenced using a high-throughput next-generation sequencing (NGS) platform able to generate million short sequences (reads) per single run. Sequences will be then processed using a bioinformatic pipeline whose steps can be summarized as follows: raw data collection, data cleaning, assembly, gene prediction, taxonomic annotation, and gene and protein abundance estimation. 19) Change in muscle histology and fibre typing. 20) Change in the muscle mitochondrial respiration. 21) Change in the muscle In vitro force characteristics. After the biopsy, fibre bundles of 4–6 mm in length and 0.5 mm in diameter will be dissected from the samples and immersed in a skinning solution to which the non-ionic detergent Brij 58 had been added. 23) Change in the muscle single fibre measurements, 24) Change in the muscle cytokine mRNA. 25) Change in the muscle redox status. 26) Change in the muscle proteomics.

A portion of the frozen muscle will be thawed on ice and prepared for proteomic analysis as

previously described. A global label-free proteomic approach will be used using an Ultimate 3000 RSLC nano system coupled to a QExactive mass spectrometer. Data analysis will be performed using Proteome Discover and Peaks7.

1) Eligibility criteria - assume men and women are included?

Inclusion Criteria:

Presence of Mild Cognitive Impairment or Mild Dementia. Recruited individuals, both men and women, will be assessed by means of Neuropsychological tests (Mini-Mental State Examination, evaluations criteria from Diagnostic and Statistical Manual for Mental Disorder-5) which will be performed by an expert Neuropsychologist.

2) Intervention description - It would be useful to know exactly how many visits the study consists of, and how many weeks the intervention will last.

Every single participant will take part in a 6-month program, including exercise sessions 3 times a week, for a total of 36 sessions.

Some important details, like when the neuropsychological test battery, or the vascular function measurements will be collected, would be useful to include here (or in another section).

Figure 1, panel B shows when all the assessments are performed, including neuropsychological investigations battery and vascular function measurements. Both these assessments are performed at T0 (right before the intervention) and at T1 (right after the 6-month intervention).

Please see the revised text in page7-8

In terms of the polyphenols contained in the chocolate, why it says "higher or equal" to 500 mg? shouldn't be exactly the same for all chocolates (i.e., 500 mg PP per 30g of chocolate)? Is this going to be standardised throughout the study, as it is quite important?

We have added clarification to page 7:

The concentrations of chocolate polyphenols are standardized and all the patients receive the same dose of chocolate product developed by Perugina Nestlè factory during the first phase of the project (before the initiation of the clinical trial) and thus of chocolate PP that were measured at the Coordinator Unit.

3) Outcomes: Please clearly specify which one is the primary outcome, fat free soft tissue mass? or change of lower limb muscle mass? You have fat free soft tissue mass in clinicaltrials.gov.

Primary Outcome Measure: Change in free-fat soft tissue mass (g). Change in free-fat soft tissue mass, (FFSTM, g), will be assessed by means of a whole-body scan on a dual-energy X-ray absorptiometry scanner. Values at the regional level (upper limbs, lower limbs, and trunk) will be also considered.

Also be more specific regarding secondary outcomes i.e., which aspects of cognitive function? A full list of secondary/tertiary/exploratory outcomes will be useful, specially as not all the ones described in the clinical and physiological assessment section are mentioned here (and clinical trials.gov)

Please see revised text in page 7-8

4) Clinical and physiological assessments - it would be useful to add a bit more info on some of the outcomes such as which neuropsychological battery will be used, or how the FMD will be measured, for example. Also, only in some cases it is reported when the measurements will be assessed (ie:"saliva

will be collected at the beginning and at the end of the 3 months treatment "...), this information should be added for all outcomes as this is very important information.

We have revised the outcome section to include these details.

5) Provision of metabolomics - which plasma will be used, before and after 3 months of treatment? which targeted analysis will be conducted?

In the revised text we included further information in page 11.

6) Allocation and blinding - which parameters will be used for the blocked randomization?

This is a randomized, double-blinded, controlled trial. A blocked randomization approach will be applied to reach the required number for each group.

Minor comments:

Page 4 line 9 - Verb missing "European older adults ARE at high risk, ..."

Current data shows that 23% of European older adults are at high risk, and 48% at some risk of PEM (1).

Page 4 Lines 19 & 20 - there is something missing in the sentence "leading to a disruption in the production of glucose, AND? reduction in energy supply to the muscles" and in the sentence starting with "The, and.."

This may be caused by an abnormal regulation of the hypothalamic-pituitary-adrenal (HPA) axis, leading to a disruption in the production of glucose, and the reduction in energy supply to the muscles. (6). The, chronically elevated cortisol level (7,8).

Page 5 line 57 - typo in "trails" (RCTs) should be trials.

Previous randomized controlled trials (RCTs)

VERSION 2 – REVIEW

REVIEWER	Rodriguez-Mateos, Ana
----------	-----------------------

	King's College London
REVIEW RETURNED	13-Oct-2023
GENERAL COMMENTS	Authors have addressed my comments.

VERSION 2 – AUTHOR RESPONSE